# GA-YOLO: A Lightweight YOLO Model for Dense and Occluded Grape Target Detection

**Jiqing Chen [1,2], Aoqiang Ma [1,\*], Lixiang Huang [1], Yousheng Su [1], Wenqu Li [1], Hongdu Zhang [1] and Zhikui Wang [1]**

[1] College of Mechatronic Engineering, Guangxi University, Nanning 530007, China
[2] Guangxi Manufacturing System and Advanced Manufacturing Technology Key Laboratory,
    Nanning 530007, China
\* Correspondence: aoqiangma@163.com; Tel.: +86-178-6888-2739

**Abstract:** Picking robots have become an important development direction of smart agriculture, and the position detection of fruit is the key to realizing robot picking. However, the existing detection models have the shortcomings of missing detection and slow detection speed when detecting dense and occluded grape targets. Meanwhile, the parameters of the existing model are too large, which makes it difficult to deploy to the mobile terminal. In this paper, a lightweight GA-YOLO model is proposed. Firstly, a new backbone network SE-CSPGhostnet is designed, which greatly reduces the parameters of the model. Secondly, an adaptively spatial feature fusion mechanism is used to address the issues of difficult detection of dense and occluded grapes. Finally, a new loss function is constructed to improve detection efficiency. In 2022, a detection experiment was carried out on the image data collected in the Bagui rural area of Guangxi Zhuang Autonomous Region, the results demonstrate that the GA-YOLO model has an mAP of 96.87%, detection speed of 55.867 FPS and parameters of 11.003 M. In comparison to the model before improvement, the GA-YOLO model has improved mAP by 3.69% and detection speed by 20.245 FPS. Additionally, the GA-YOLO model has reduced parameters by 82.79%. GA-YOLO model not only improves the detection accuracy of dense and occluded targets but also lessens model parameters and accelerates detection speed.

**Keywords:** picking robot; computer vision; grape detection; GA-YOLO; dense and occluded target; lightweight model

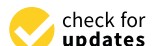



## 1. Introduction

Grapes, known as the queen of fruits, have high economic value. The short fruit period of grapes means that timely picking is essential for quality. Currently, hand grape harvesting is the most common method, which takes a lot of time and labor. With the transfer of rural labor from agriculture to non-agricultural industries, the rural surplus labor is gradually decreasing [1]. Therefore, developing grape-picking robots has important research prospects. At present, picking robots mainly rely on the vision system to realize the location of fruits. Accurately detecting the location of the fruit is the key to achieving picking [2]. Especially in the complex environment of grape orchards, is disturbed by factors such as illumination change, leaf occlusion, and fruit overlapping, which bring huge challenges to picking robots.

Traditional fruit detection methods, such as support vector machine [3], template matching [4], edge detection [5], and threshold segmentation [6], mainly extract inherent features, such as geometric shape [7], color [8–10], spectral information [11], texture [12] and edge [13], to realize the detection of the grape region. Liu et al. [14] used the least square method to fit the elliptic boundary of pomelo, to realize the segmentation of pomelo. Lin et al. [4] proposed a local template matching algorithm and trained a new vector machine classifier by using color and texture, which can detect tomatoes, pumpkins, mangoes, and oranges. Nazari et al. [15] designed an RGB classifier based on the color

difference between red grapes and the background, which can segment red grapes. Pérez-Zavala et al. [16] extracted the edge gradient information and surface texture information of grapes as classification features and used the support vector machine classifier to realize the segmentation of grapes and the background. Behroozi-Khazaei et al. [8] put forward a method combining an artificial neural network and genetic algorithms, which can overcome the problem that greens grapes are similar to the background. Traditional grape detection methods can achieve good segmentation results when only a few fruits with a specified color and shape. At the same time, traditional image processing techniques rely on high-quality images and require complex artificial features. However, when there are complex scenes, such as scenes with changing illumination, scenes with dense fruits, and scenes with hidden fruits, the performance of fruit detection becomes poor. Under the circumstances, multiple overlapping grapes may be detected as one.

In the recent ten years, with the wide application of deep learning, great breakthroughs have been made in the object detection field [17–22]. Gao et al. [23] divided the blocked apples into three categories, including apples occluded by leaves, apples occluded by branches, and apples occluded by other apples, and used the Faster R-CNN algorithm to detect the occluded apples. Tu et al. [24] proposed a multi-scale feature fusion MS-FRCNN algorithm, which combined the semantic information of the deep network and the location information of the shallow network to improve the detection accuracy in the case of dense passion fruit. Mai et al. [25] increased the single classifier in Faster-RCNN to three classifiers, which effectively enhanced the detection performance of dense fruit targets. Ding et al. [26] improved the SSD model by using the receptive field block and attention mechanism, which effectively reduced the missed detection rate of occluded apples. Behera et al. [27] changed IOU to MIOU in the loss function of Fast RCNN, which improved the recognition performance of occluded and dense fruits. Tu et al. [24] and Ding et al. [26] improved the feature fusion module of the model, and Behera et al. [27] improved the loss function to solve the issue of difficult recognition of occluded and dense targets. However, due to the slow detection speed and a large number of parameters, the above models are difficult to deploy on the mobile end of harvesting robots.

In order to solve the issues of large parameters and slow detection speed, some scholars have studied in the field of lightweight. Generally speaking, the detection speed increases with the decrease in the model parameters. The main methods to reduce the parameters are replacing the convolution module and reducing the convolution layer [28–31]. Mao et al. [32] proposed the Mini-YOLOv3 model, which used depthwise separable convolution and point group convolution to decrease the parameters. A lightweight YOLOv4 model was proposed by Zhang et al. [33], the backbone network Darknet-53 of YOLOv4 is replaced with the GhostNet network and the basic convolution is replaced with a depthwise separable convolution in the neck and head. Ji et al. [34] took YOLOVX-Tiny as the baseline, adopted a lightweight backbone network, and proposed a method for apple detection based on Shufflenetv2-YOLOX. Fu et al. [35] used $1 \times 1$ convolution to decrease the parameters of the original model and proposed the DY3TNet model to detect kiwifruit. Li et al. [36] reduced the calculations and parameters by introducing deep separable convolution and ghost modules. Liu et al. [37] proposed the YOLOX-RA model, which pruned part of the network structure in the backbone network and used depth separable convolution in the neck network. Cui et al. [38] changed the backbone network from CSPdarknet-Tiny to ShuffleNet in YOLOv4-tiny and reduced the three detection heads to one detection head. Zeng et al. [39] replaced CSPdarknet with Mobilenetv3 and compressed the neck network of YOLOv5s by pruning technology [40].

Although these models achieve lightweight, the detection accuracy suffers. In the vineyard, clusters of grapes grow densely and overlap each other, and the huge leaves easily cover the grapes. The complex growing environment leads to a low recall rate of the deep learning detection model for grape detection. In addition, the model parameters with high detection accuracy are redundant, which makes it difficult to deploy to the mobile end of the picking robots. The existing detection model can hardly meet the two advantages of

detection accuracy and detection speed. To sum up, our research objective is to solve the problem that targets are difficult to identify while ensuring the accuracy of model detection and reducing the parameters of the model. In this paper, a GA-YOLO model with fast detection speed, small parameters, and a low missed detection rate is proposed for dense and occluded grapes.

In short, our innovations are as follows:

(1) A new backbone network SE-CSPGhostnet is designed, which greatly reduces the parameters.
(2) ASFF mechanism is used to address the issues of difficult detection of occluded and dense targets, and the model's detection accuracy is raised.
(3) A novel loss function is constructed to improve detection efficiency.

The architecture of this paper is as follows: Section 1 introduces the background, significance, and current status. Section 2 introduces dataset collection, annotation, and augmentation. Section 3 introduces the GA-YOLO algorithm. Section 4 contains the experimental process, the comparison of model performance, and the analysis of the results. Section 5 describes the use of human–computer interaction interface. Section 6 discusses the experimental results and points out the limitations of the algorithm. Section 7 concludes the paper and provides future research plans.

In the paper, the full names and acronyms are displayed in Table 1.

**Table 1.** The acronyms and full names.

| Acronyms | Full Name |
| --- | --- |
| ASFF | Adaptively Spatial Feature Fusion |
| CBL | Convolution, Batch normalization and Leaky Relu activation function |
| CBM | Convolution, Batch normalization and Mish activation function |
| CSP | Cross Stage Partial |
| FLOPs | Floating point operations per second |
| FPNet | Feature Pyramid Network |
| FPS | Frames Per Second |
| GBM | Ghost convolution, Batch normalization and Mish activation functions |
| IOU | Intersection over Union |
| mAP | Mean Average Precision |
| PANet | Path Aggregation Network |
| RCNN | Regions with CNN features |
| Res element | Residual element |
| SENet | Squeeze-and-Excitation Networks |
| SPP | Spatial Pyramid Pooling |
| SSD | Single Shot MultiBox Detector |
| YOLO | You Only Look Once |

## 2. Datasets

### 2.1. Collection of Datasets

The study's grape datasets were collected from 21 June 2022 to 26 June 2022 in Bagui Garden, Nanning City, Guangxi Zhuang Autonomous Region, including three varieties of grapes: "Kyoho", "Victoria" and "Red Fuji". We used Daheng Industrial Camera MER-132-43U3C-L for the acquisition of datasets. All images were acquired under natural lighting at 8:30 am, 11:30 noon, 2:30 pm, and 5:30 pm on sunny and overcast days. The distance from the camera lens to the grapes is 0.5 m~1.2 m. The camera's shooting angles include flat, up, and down. The camera is shown in Figure 1, and the basic parameters of the industrial camera are displayed in Table 2.

In order to avoid the overfitting of the network model caused by the single feature of the datasets, 200 "Kyoho Grapes" images, 200 "Red Fuji Grapes" images, and 200 "Victoria Grapes" images were collected in consideration of different light intensity, different occlusion degree, and different fruit sparseness. Figure 2 shows the im-

ages of three grape varieties, and Table 3 shows the number of grape images in different collection conditions.

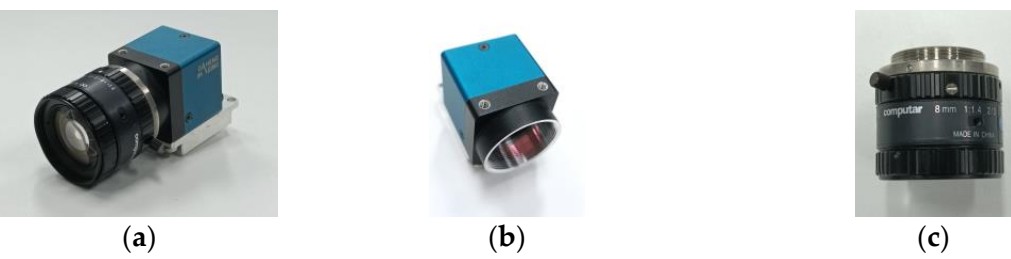

(**a**)  (**b**)  (**c**)

**Figure 1.** Daheng Industrial Camera: (**a**) overall camera; (**b**) camera base; (**c**) camera lens.

**Table 2.** The basic parameters of the industrial camera.

| Parameter | Value |
| --- | --- |
| Model | MER-132-30UC |
| Frame rate | 30 fps |
| Sensor type | 1/3″ CCD |
| Spectrum | black/color |
| Data Interface | USB2.0 |
| Working temperature | 0–60 °C |
| Working humidity | 10–80% |
| Resolution ratio | 1292 × 964 |

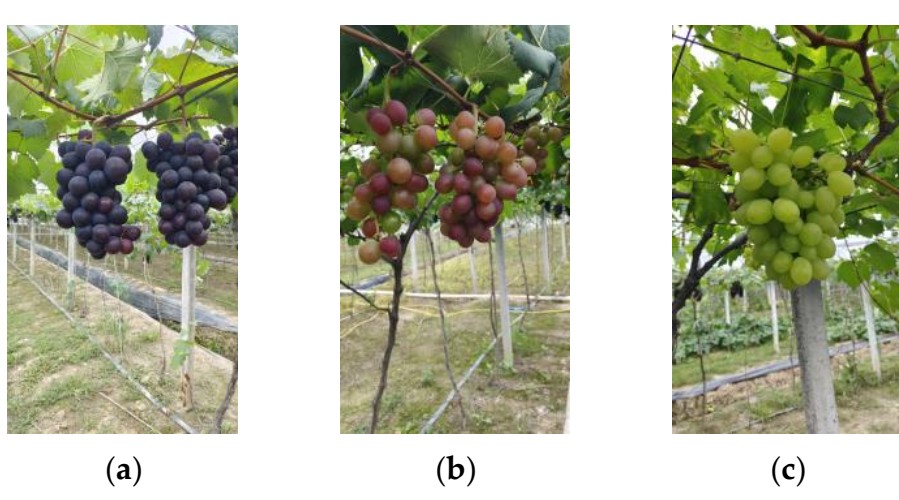

(**a**)  (**b**)  (**c**)

**Figure 2.** Three grape varieties: (**a**) Kyoho; (**b**) Red Fuji; (**c**) Victoria.

**Table 3.** Number of grapes images in different collection conditions. ("√" indicates "yes"; "×" indicates "no").

| Varieties | Kyoho | | | | Red Fuji | | | | Victoria | | | |
| --- | --- | --- | --- | --- | --- | --- | --- | --- | --- | --- | --- | --- |
| Is the number of grape clusters more than twelve? | √ | | × | | √ | | × | | √ | | × | |
| Is there occlusion? | √ | × | √ | × | √ | × | √ | × | √ | × | √ | × |
| Number of images | 100 | 100 | 50 | 50 | 50 | 50 | 100 | 100 | 50 | 50 | 50 | 50 |

### 2.2. Annotation of Datasets

This paper uses labelImg software [41] for labeling, annotation format is Pascal VOC. LabelImg software is shown in Figure 3a, and the label format corresponding to the labeled picture is shown in Figure 3b. Image labeling is based on the following principles:

(1) unripe grapes are not labeled; (2) grapes falling on the ground will not be labeled; (3) grapes with an occlusion area exceeding $\frac{4}{5}$ are not labeled; (4) grape images that are blurred, but the grape area is larger, it is also labeled; (5) when labeling, ensure that the label box and the grape area overlap to the maximum.

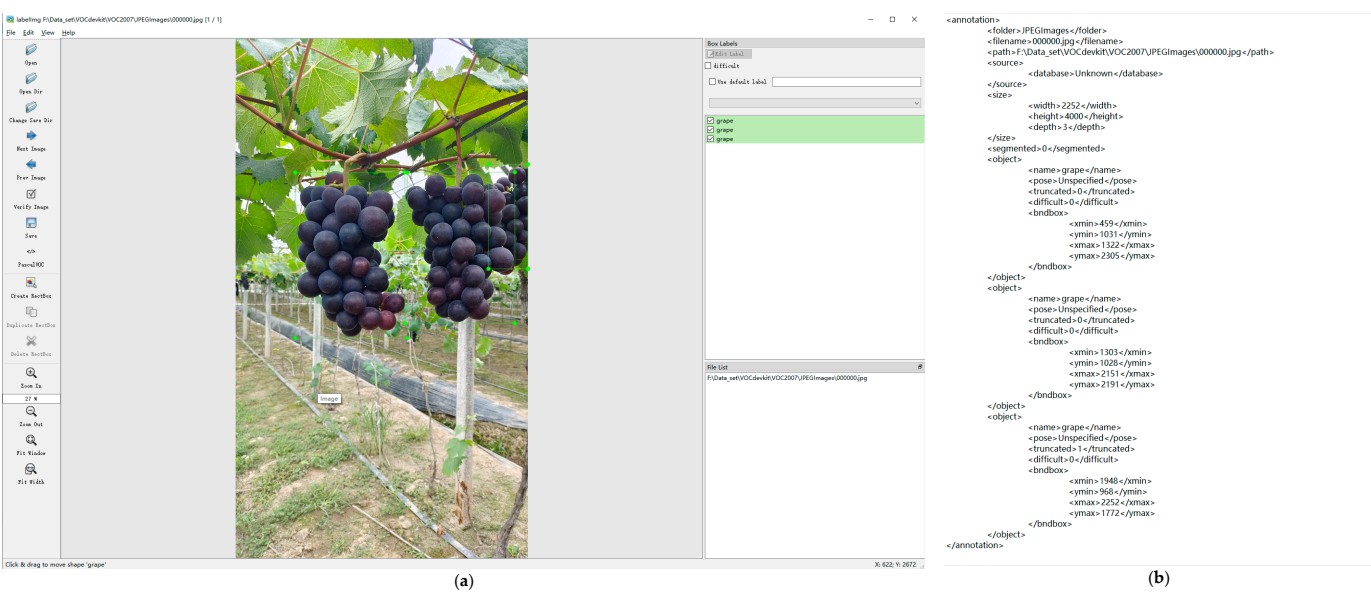

(a) | (b)

**Figure 3.** (**a**) LabelImg software; (**b**) XML label file.

### 2.3. Augmentation of Datasets

Data augmentation has the advantages of saving time for making labels, preventing model overfitting, and improving model generalization ability. The augmentation methods are shown in Figure 4, which contains 14 augmentation methods such as scaling, cropping, rotation, brightness change, saturation change, contrast change, blurring process, and mosaic data augmentation. Finally, $600 \times 14 = 8400$ valid images are obtained. According to the ratio of 7:2:1, the photos are separated into training set, validation set, and test set.

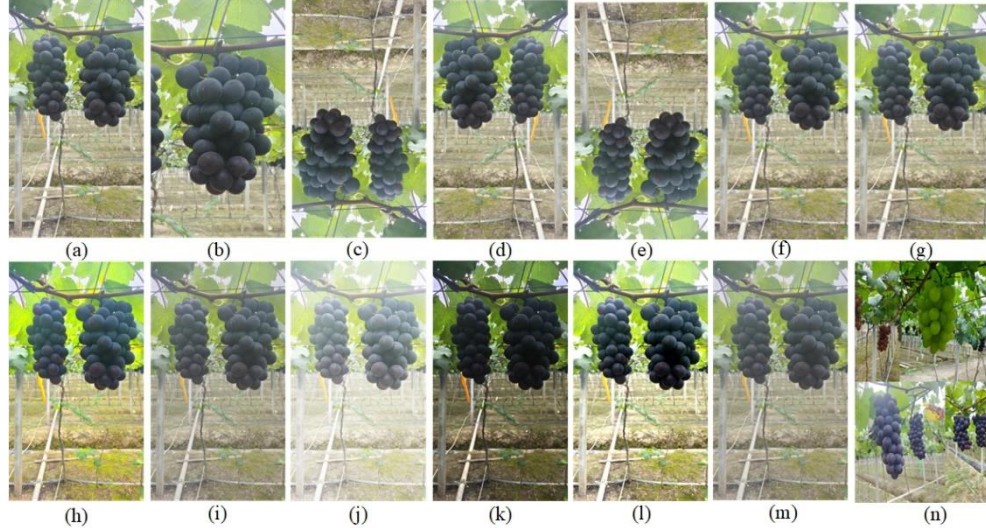

**Figure 4.** Image augmentation method: (**a**) original image; (**b**) crop and zoom; (**c**) rotate by 180 degrees; (**d**) flip horizontally; (**e**) vertical turnover; (**f**) fuzzy median value; (**g**) Gaussian blur; (**h**) 50% increase in saturation; (**i**) 50% reduction in saturation; (**j**) 50% increase in brightness; (**k**) 50% reduction in brightness; (**l**) 50% increase in contrast; (**m**) 50% reduction in contrast; (**n**) mosaic data enhancement method.

## 3. Methodologies

### 3.1. YOLOv4 and GA-YOLO

The YOLOv4 model's structure is shown in Figure 5a. The GA-YOLO model is improved on the basis of the YOLOv4 [22], as shown in Figure 5b. We propose a new backbone network SE-CSPGhostnet and incorporate the ASFF mechanism into the head network. Furthermore, a new loss function is used to improve the detection performance.

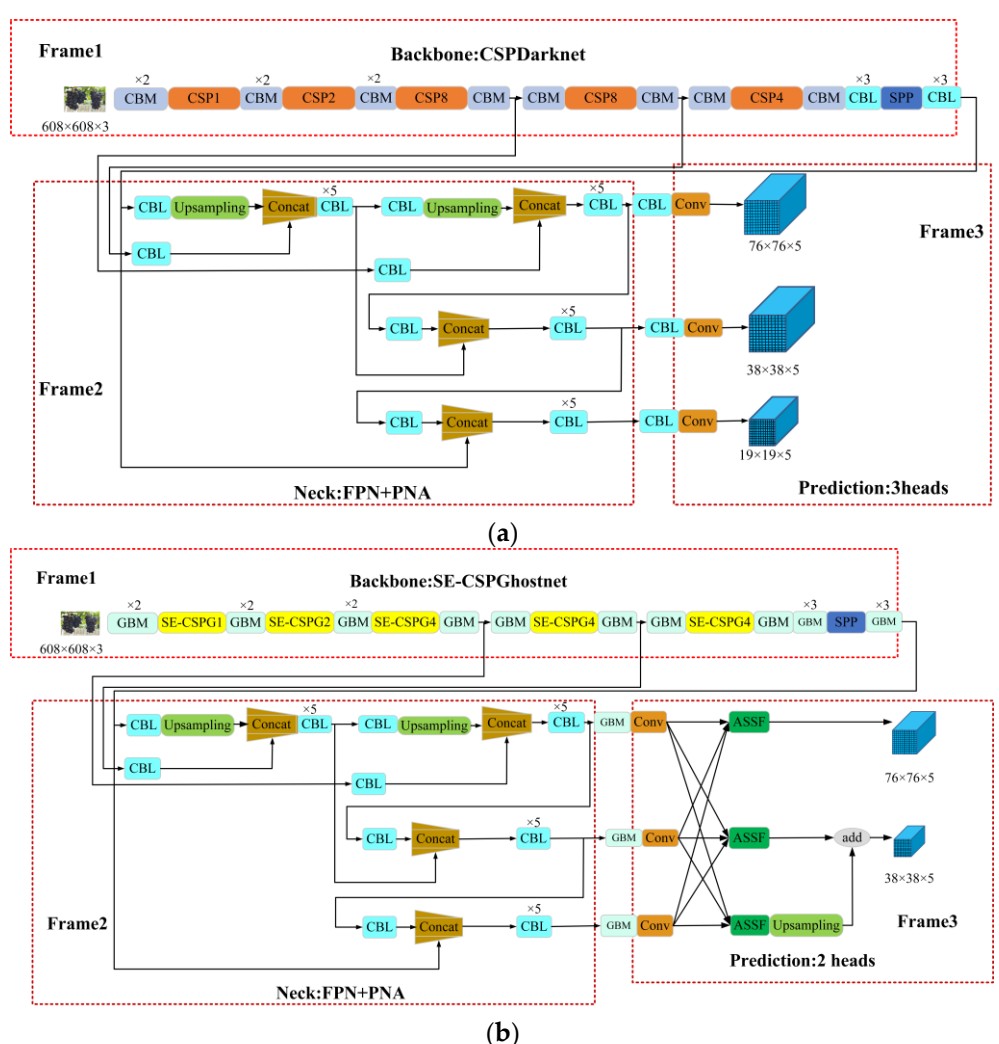

**Figure 5.** (**a**) YOLOV4 network model structure (**b**) GA-YOLO network model structure.

In Figure 5a, the YOLOv4 network model consists of modules such as SPP, CBM, CBL, and CSP. Among them, Spatial Pyramid Pooling [42] (SPP) fixes feature maps of any size as feature vectors of the same length through a pooling of three scales. CBL contains convolution, Batch normalization, and Leaky Relu activation functions, which are used in the latter position of the backbone network to extract features. The CBM module is composed of convolution, Batch normalization, and Mish activation functions, which are used in the front position of the backbone network to extract features. We changed the convolution in CBM to ghost convolution and proposed the GBM module. In order to reduce parameters of model, the CBM and CBL modules in the backbone network are changed to GBM modules.

Meanwhile, the CBL modules at the junction of the neck network and the head network were changed to GBM modules. CBM, CBL, GBM, and SPP are shown in Figure 6.

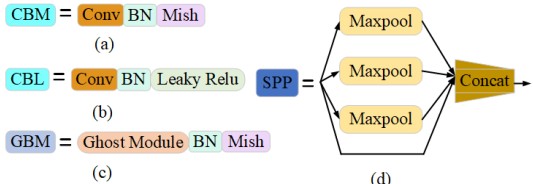

**Figure 6.** (**a**) CBM module (**b**) CBL module (**c**) GBM module (**d**) SPP module.

The CBM uses ordinary convolution, and convolution process is shown in Figure 7a. GBM uses ghost convolution [28], which greatly reduces the number of parameters, as shown in Figure 7b. The primary distinction between ghost convolution and ordinary convolution is that ghost convolution has two convolution processes. Firstly, $\frac{n}{s}$ intermediate feature maps are obtained by ordinary convolution. Then, the intermediate feature map is convoluted with a convolution kernel of $d \times d$ size to obtain $(s-1) \times \frac{n}{s}$ feature maps. Finally, the $\frac{n}{s}$ intermediate feature maps acquired in the first step and the $(s-1) \times \frac{n}{s}$ feature maps acquired in the second step are superimposed on the channel dimension to obtain a total of n feature maps. The parameter quantity of GBM is shown in Formula (1). In contrast, the ordinary convolution in Figure 7a is a direct convolution to obtain n output feature maps. The parameter quantity of CBM is shown in Formula (2). Obviously, the amount of final feature maps of GBM convolution and ordinary convolution is the same. The ratio of parameters of CBM and GBM is shown in Formula (3). Through calculation and analysis, theoretically, the parameter quantity of GBM is $\frac{1}{s}$ of that of CBM.

$$P_1 = h' \times w' \times \frac{n}{s} \times k \times k \times c + (s-1) \times h' \times w' \times \frac{n}{s} \times d \times d \tag{1}$$

$$P_2 = h' \times w' \times n \times k \times k \times c \tag{2}$$

$$r_s = \frac{P_2}{P_1} \approx s \tag{3}$$

where, $h'$ represents the length of output feature map; $w'$ represents the width of output feature map; $n$ represents the number of channels for output feature map; $k$ represents the size of the convolution kernel; $c$ represents the channel number of convolution kernel; $s$ represents the ratio of the number of channels of output feature map to the number of channels of input feature map; $d$ represents the size of convolution kernel in the second convolution of ghost convolution.

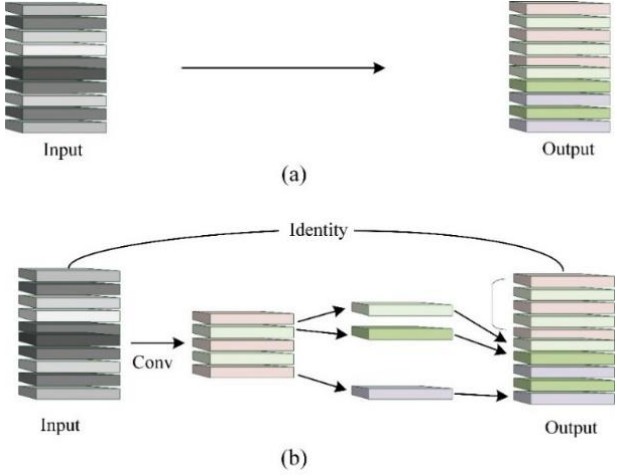

**Figure 7.** Two convolution methods: (**a**) ordinary convolution (**b**) ghost convolution.

### 3.2. Improvement of GA-YOLO Backbone Network

The role of backbone network is to extract features, and its structure is shown in Figure 5. We made three improvements to YOLOv4 backbone network CSPdarknet to obtain the GA-YOLO backbone network CSPGhostnet: (1) change the CBM and CBL to GBM; (2) change the CSP structure to SE-CSPG structure; (3) reduce the number of iterations of the SE-CSPG module. The above three improved methods all greatly reduce parameters and calculations.

The CSP structure [43] of CSPdarknet has a Res unit component that iterates X times, as shown in Figure 8a. In Res unit, the CBM module is replaced with a GBM module. At the same time, the insertion of Squeeze-and-Excitation Networks (SENet) improves the performance of the Res element to solve the problem of gradient degradation. After adding skip connection and Res element in SE-CSPG, the shallow feature information is integrated into the deep feature information, which improves the generalization performance of the model.

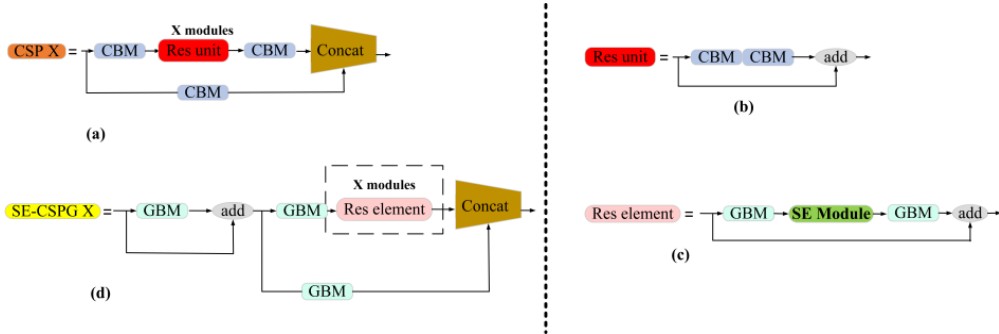

**Figure 8.** (**a**) CSP structure (**b**) Res unit structure (**c**) SE-CSPG structure (**d**) Res element structure.

The attention mechanism can correct the features, make the network focus on important local information. Useless feature information is filtered out, so as to simplify the model and accelerate the calculation. SENet [44] mainly studies the relationship between channels and realizes the effect of adaptively correcting channel characteristics. SENet is a typical representative of the channel attention mechanism, as shown in Figure 9a.

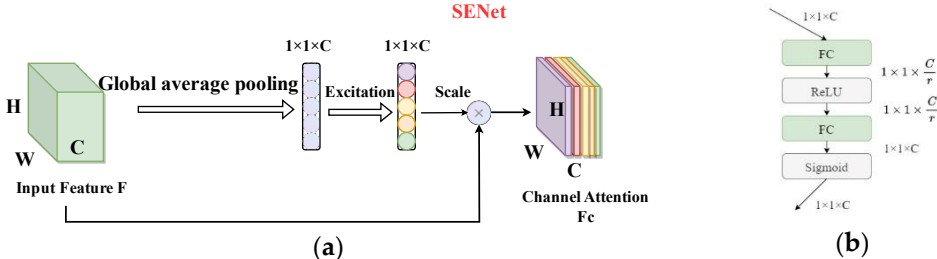

**Figure 9.** (**a**) SENet (**b**) the process of Excitation.

The input feature map $F$ compresses the two-dimensional feature $H \times W$ of each channel into a real number through global average pooling. After this compression operation, the size of the feature map is converted from the original $H \times W \times C$ to $1 \times 1 \times C$. The Excitation operation is performed on the obtained feature map $F_g$ to generate a weight value for each feature channel. The Excitation operation is to use two fully connected layers to build the correlation between channels, as shown in Figure 9b. The normalized weight is obtained after the activation function Sigmoid. The $F_e$ represents the importance of the channel, and it is weighted to the features of each channel in the $F$ to obtain the channel attention feature map $F_c$.

### 3.3. Introduction of GA-YOLO Neck Network

The neck network is shown in Frame 2 of Figure 5. The role of the neck network is to fuse the features of different feature layers. Feature Pyramid Networks (FPN) [45] and Path Aggregation Networks [46] (PAN) are used as the feature fusion module, making full use of the semantic information of high-dimensional feature maps and the location information of low-dimensional feature maps. The feature fusion of neck network improves the detection accuracy of dense and occluded targets.

### 3.4. Improvement of GA-YOLO Head Network

The structure of the head network is shown in Frame 3 of Figure 5. The role of the head network is to predict the class and location. In the head network, Adaptively Spatial Feature Fusion [47] (ASFF) is added to the front of the prediction head. ASFF can adaptively learn the spatial weight of each scale feature map fusion, which is used to solve the problem of inconsistent scales in spatial feature fusion. The structure of ASFF is shown in Figure 10.

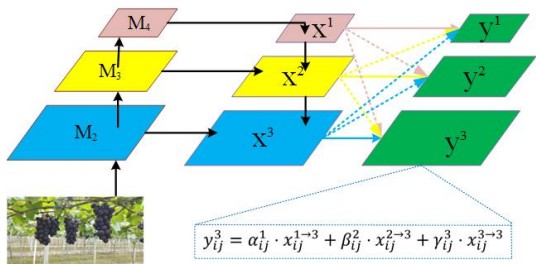

**Figure 10.** ASFF structure diagram.

Where, $M_1$, $M_2$ and $M_3$ represent the feature map obtained by convolution of the backbone network; $x^1$, $x^2$ and $x^3$ represent the feature map after PANet feature fusion; $y^1$, $y^2$ and $y^3$ represent the feature maps after ASFF mechanism processing.

In Figure 10, the learning process of spatial weight of ASFF is shown in Formula (4). Where, $x^l$ represents the $l$-layer feature map and $x_{ij}^{s \to t}$ represents the feature map from $s$-layer resize to $t$-layer. $\alpha_{ij}^l, \beta_{ij}^l, \gamma_{ij}^l$ are the learned spatial weights, which represent the importance of the pixel at the coordinate $(i, j)$ in the feature map of the $l$th layer. Meanwhile, $\alpha_{ij}^l, \beta_{ij}^l, \gamma_{ij}^l$ satisfy $\alpha_{ij}^l + \beta_{ij}^l + \gamma_{ij}^l = 1$ and $\alpha_{ij}^l, \beta_{ij}^l, \gamma_{ij}^l \in [0, 1]$. $\alpha_{ij}^l, \beta_{ij}^l, \gamma_{ij}^l$ can be obtained according to Formula (5). In Formula (5), $\lambda_\alpha^l, \lambda_\beta^l, \lambda_\gamma^l$ are control parameters, which can be learned by back-propagation of the network.

$$y_{ij}^3 = \alpha_{ij}^1 \cdot x_{ij}^{1 \to 3} + \beta_{ij}^2 \cdot x_{ij}^{2 \to 3} + \gamma_{ij}^3 \cdot x_{ij}^{3 \to 3} \tag{4}$$

$$\alpha_{ij}^l = \frac{e^{\lambda_{\alpha_{ij}}^l}}{e^{\lambda_{\alpha_{ij}}^l} + e^{\lambda_{\beta_{ij}}^l} + e^{\lambda_{\gamma_{ij}}^l}} \tag{5}$$

### 3.5. Improvement of GA-YOLO Loss Function

The YOLOv4 loss function before improvement includes three sections: confidence loss, rectangular box loss and classification loss, as shown in Formula (6).

$$loss = a \cdot loss_{conf} + b \cdot loss_{box} + c \cdot loss_{clc} \tag{6}$$

Three improvements to the loss function are made: (1) Since there is only one class of target to be detected, set $c = 0$, which removes the ineffective classification loss. (2) Increase the weight of confidence loss, take $a = 0.6$, $b = 0.4$. (3) Confidence loss consists of target confidence loss and background confidence loss. The weight of target confidence loss

is increased, taking $\lambda_{obj} = 0.7$, $\lambda_{noobj} = 0.3$. The loss of improved confidence is shown in Formula (7).

$$loss_{conf} = \lambda_{obj} \sum_{i=0}^{S^2} \sum_{j=0}^{B} 1_{i,j}^{obj} [\hat{C}_i^j \log(C_i^j) + (1 - \hat{C}_i^j) \log(1 - C_i^j)]$$
$$+ \lambda_{noobj} \sum_{i=0}^{S^2} \sum_{j=0}^{B} 1_{i,j}^{noobj} [\hat{C}_i^j \log(C_i^j) + (1 - \hat{C}_i^j) \log(1 - C_i^j)] \tag{7}$$

In Formulas (6) and (7), some proper nouns are defined as follows: confidence indicates the confidence degree of the predicted rectangular box containing the target, and binary cross entropy loss is employed for calculating confidence loss. $S^2$ indicates the number of divided grids in image, $B$ indicates the number of prior frames in each grid, $\lambda_{obj}$ indicates the weight factor of the target's confidence loss and $\lambda_{noobj}$ indicates the weight factor of the background's confidence loss. $\hat{C}_i^j$ is the label value of the prediction box's confidence and $C_i^j$ is the predicted value of the prediction box's confidence. $1_{i,j}^{obj}$ indicates that if the ith grid's $j$th prediction box has a target, its value is 1, otherwise, it is 0. $1_{i,j}^{noobj}$ indicates that there is no target in the ith grid's $j$th prediction box, and its value is 1, otherwise it is 0.

In Formula (6), rectangular box loss is employed to calculate the position error between the predicted box and the ground-truth box, including the error loss of the central point coordinate and the height and width of the rectangular box, which is calculated by using CIOU loss function [48], as shown in Formula (8).

$$loss_{box} = \sum_{i=0}^{S^2} \sum_{j=0}^{B} 1_{i,j}^{obj} [1 - IOU + \frac{\rho^2}{c^2} + \frac{\frac{16}{\pi^4}\left(arctan\frac{w^{gt}}{h^{gt}} - arctan\frac{w}{h}\right)^4}{1 - IOU + \frac{4}{\pi^2}\left(arctan\frac{w^{gt}}{h^{gt}} - arctan\frac{w}{h}\right)^2}] \tag{8}$$

In Formula (8), *IOU* represents the ratio of the area of the intersection region between ground-truth rectangular box and predicted rectangular box to the area of the merged region. $\rho$ represents the distance between the central point of prediction rectangular box and the central point of ground-truth rectangular box. $c$ represents the length of the diagonal of the external rectangular box of the prediction rectangular box and the ground-truth rectangular box. $w^{gt}$ and $h^{gt}$ represent the width and height of the ground-truth rectangular box. $w$ and $h$ represent the width and height of the prediction rectangular box.

## 4. Results of Experiment

### 4.1. Experimental Details

In order to confirm that the GA-YOLO substantially improve the detection performance, 8400 grape datasets are used to conduct experiments. The experimental hardware and software configuration parameters are displayed in Table 4.

**Table 4.** The basic parameters of the industrial camera.

| Hardware/Software | Configuration/Version |
|---|---|
| CPU | Intel(R) Xeon(R) CPU E5-2680 |
| GPU | Tesla M40 24 G $\times$ 4 |
| Memory | DDR4 64G KF3200C16D4/8GX |
| Hard disk | SSD 980 500 G |
| Operating system | Ubuntu20.04.1 |
| Python | 3.9 |
| Pytorch | 1.8.1 |
| CUDA | 10.0.3 |

To ensure the fairness of the experimental comparisons, all models are trained under the same hardware condition and the same initial training parameters. The learning rate is adopted by means of the cosine annealing decay method. The weights are saved every

10 epochs during the training process. The specific training initial parameters are displayed in Table 5.

**Table 5.** Initial training parameters.

| Parameter | Form/Value |
|---|---|
| Init-learning rate | 0.01 |
| Min-learning rate | Init-lr $\times$ 0.01 |
| Optimizer-class | SGD |
| momentum | 0.937 |
| Lr-decay-class | Cos |
| Weight decay | 0.0005 |
| Num-works | 4 |
| Batch size | 64 |
| Epoch | 50 |

*4.2. Metrics for Evaluation*

FPS refers to the number of images that can be detected per second, which is employed to assess the network model's detecting speed. Parameters represent the volume of parameters that require training in the network model. Weights represent the size of the weight file obtained by the final training of the network model. Parameters and weights are employed to evaluate the size of the network model, and the size of the weights is generally four times the size of the parameters. The smaller the parameters and weights, the easier the model to be deployed to the mobile terminal of the picking robot. Floating-point operations per second (FLOPs) are employed to evaluate the calculation effort of the model. The precision rate, recall rate, $F_1$ score, and *AP* are employed to evaluate the accuracy of the target detection method.

The precision rate indicates the ratio of being a positive sample among predicted positive samples, as shown in Formula (9):

$$Precision = \frac{TP}{TP + FP} \times 100\% \tag{9}$$

The *recall* rate indicates the ratio of correctly predicted positive samples to labeled positive samples, as shown in Formula (10):

$$Recall = \frac{TP}{TP + FN} \times 100\% \tag{10}$$

The harmonic mean of the *precision* rate and *recall* rate is the $F_1$ score, as shown in Formula (11):

$$F_1 = \frac{2 \times Precision \times Recall}{Precision + Recall} \tag{11}$$

The two indicators of precision rate and recall rate show a negative correlation. Therefore, to comprehensively assess the quality of the algorithm, the PR curve is usually drawn with the recall rate as the horizontal axis and with the precision rate as the vertical axis. The area below the PR curve is average precision (*AP*) value, as shown in Formula (12):

$$AP = \int_0^1 p(r)dr \tag{12}$$

*4.3. Comparison of Network Models*

4.3.1. Calculation Volume, Parameter Volume, and the Size of Weight File

The GA-YOLO network model is compared with mainstream detection network models such as Faster RCNN, YOLOv3, YOLOv4, SSD, YOLOv4-MobileNetv1, YOLOv4-MobileNetv2, YOLOv4-MobileNetv3, YOLOv4-tiny, YOLOv5s, YOLOv5m, YOLOv5l, and YOLOv5x.

On account of the large number of network models engaged in the comparison, the models are divided into the light network model (0 < FLOPs ≤ 50 G), medium network model (50 < FLOPs ≤ 100 G), and large network model (FLOPs > 100 G) according to the calculation volume of the network model. Therefore, in Table 6, the light network model contains YOLOv4-MobileNetv1, YOLOv4-MobileNetv2, YOLOv4-MobileNetv3, YOLOv4-tiny, GA-YOLO, and YOLOv5s, the medium network model contains YOLOv3, YOLOv4, and YOLOv5m, and the large network model contains Faster RCNN, SSD, YOLOv5l, and YOLOv5x. The comparison results on calculation volume, parameter volume, and the size of the weight file are displayed in Table 6.

**Table 6.** Comparison of calculation volume, parameter volume, and weight file of different network models.

| Network Models | FLOPs (G) | Parameters (M) | Weights (M) |
|---|---|---|---|
| Faster RCNN | 252.676 | 136.689 | 108 |
| SSD | 115.513 | 23.612 | 90.6 |
| YOLOv3 | 65.520 | 61.524 | 235 |
| YOLOv4 | 59.7758 | 63.938 | 244 |
| YOLOv4-MobileNetv1 | 21.285 | 14.267 | 57.1 |
| YOLOv4-MobileNetv2 | 16.185 | 12.376 | 49.4 |
| YOLOv4-MobileNetv3 | 14.999 | 11.304 | 53.6 |
| YOLOv4-tiny | 16.438 | 7.057 | 28.4 |
| GA-YOLO | 13.860 | 11.003 | 32.5 |
| YOLOv5s | 16.377 | 8.064 | 32.1 |
| YOLOv5m | 50.404 | 21.056 | 80.6 |
| YOLOv5l | 114.240 | 46.631 | 178 |
| YOLOv5x | 217.323 | 87.244 | 333 |
| Faster RCNN | 252.676 | 136.689 | 108 |

The data are analyzed in Table 6. Firstly, the calculation volume, parameter volume and weight file size of the GA-YOLO model are 13.860 G, 11.003 M, and 32.5 M, respectively, which are 76.81%, 82.79%, and 86.68% lower than YOLOv4. Secondly, GA-YOLO is 34.88%, 14.37%, 7.59%, 15.68%, and 15.68% lower in calculation volume than light networks such as YOLOv4-MobileNetv1, YOLOv4-MobileNetv2, YOLOv4-MobileNetv3, YOLOv4-tiny, and YOLOv5s, respectively. At the same time, GA-YOLO is 22.88%, 11.09%, and 2.66%, lower in parameter volume than light networks such as YOLOv4-MobileNetv1, YOLOv4-MobileNetv2, and YOLOv4-MobileNetv3, respectively. Again, GA-YOLO is at least 78.85%, 47.74%, and 59.68% lower than medium-sized networks such as YOLOv3, YOLOv4, and YOLOv5m on calculation volume, parameter volume and the size of the weight file. Finally, GA-YOLO is at least 87.87%, 53.40%, and 64.13% lower than large networks such as Faster RCNN, SSD, YOLOv5l, and YOLOv5x on GFLOPs, params, and weights. This shows that the use of ghost convolution greatly decreases the volume of parameters and calculations.

### 4.3.2. Comparison of Convergence Speed

In order to confirm that the training convergence speed of the GA-YOLO has been improved after being lightweight, it is compared with other network models in Table 6. The loss value of each epoch of training is recorded, and the loss value change graph is drawn, as shown in Figure 11. Where the horizontal and vertical axes are epoch and loss values, respectively. For the convenience of comparison, we draw network models with similar convergence speed and loss value in one graph. The statistics of the convergent algebra are shown in Table 7.

According to Figure 10 and Table 7, the SSD has the slowest convergence speed, and it converges at the 50th epoch. The convergence speed of Faster RCNN and YOLOv5s is also relatively slow, reaching convergence after the 35th epoch and 40th epoch, respectively. YOLOv4-MobileNetv1, YOLOv4-MobileNetv2, and YOLOv4-MobileNetv3 have basically the same convergence speed, and they all reach convergence around the 30th epoch. YOLOv4-tiny has the fastest convergence speed and has basically converged in the 12th

epoch. GA-YOLO completed the convergence at the 15th epoch, which is about 7 epochs faster than the YOLOv4 network before the improvement, which shows that GA-YOLO saves the training time of the model.

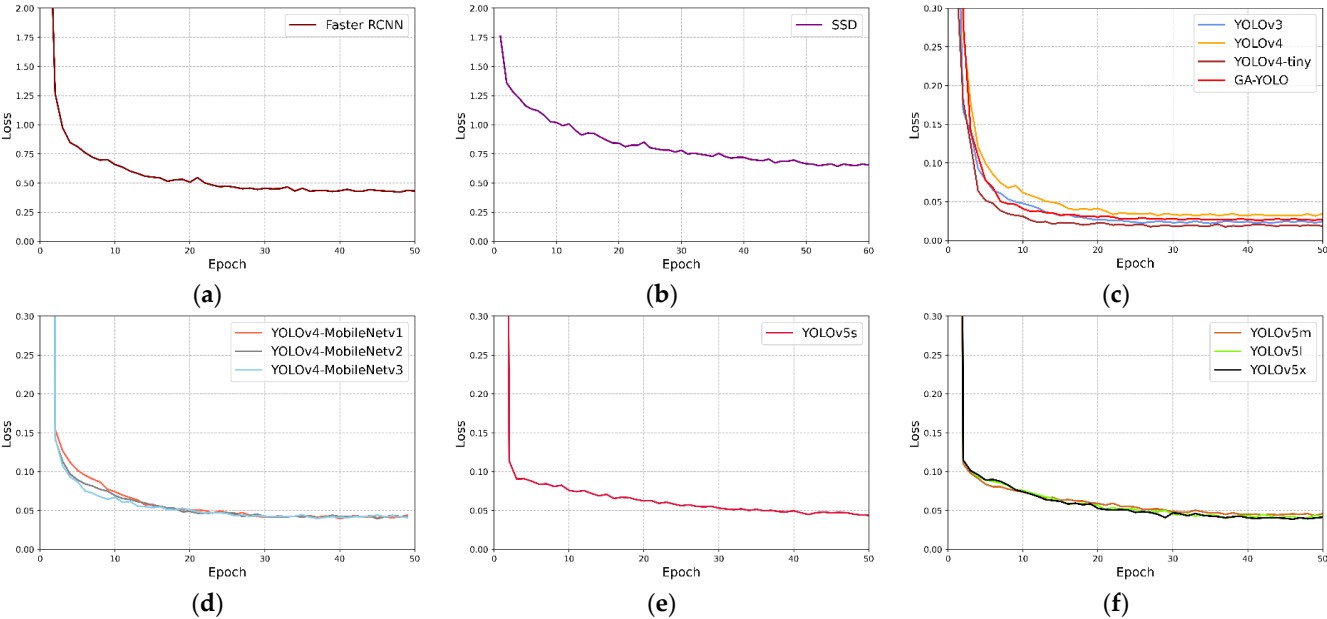

**Figure 11.** Loss convergence graphs of different network models. (**a**) Loss convergence graph of Faster RCNN; (**b**) Loss convergence graph of SSD; (**c**) Loss convergence graphs of YOLOv3, YOLOv4, YOLOv4-tiny and GA-YOLO; (**d**) Loss convergence graphs of YOLOv4-MobileNetv1, YOLOv4-MobileNetv2 and YOLOv4-MobileNetv3; (**e**) Loss convergence graphs of YOLOv5s; (**f**) Loss convergence graphs of YOLOv5m, YOLOv5l and YOLOv5x.

**Table 7.** Convergence epochs of different network models.

| Network Models | Epoch | Network Models | Epoch |
|---|---|---|---|
| Faster RCNN | 35 | SSD | 50 |
| YOLOv3 | 25 | YOLOv4 | 22 |
| YOLOv4-tiny | 13 | GA-YOLO | 15 |
| YOLOv5s | 40 | YOLOv5m | 32 |
| YOLOv5l | 33 | YOLOv5x | 35 |
| YOLOv4-MobileNetv1 | 30 | YOLOv4-MobileNetv2 | 30 |
| YOLOv4-MobileNetv3 | 30 | | |

### 4.3.3. Ablation Experiment

The ablation experiment aims to verify the role played by the SE-CSPGhostnet backbone network module, ASFF module, and improved loss function in the GA-YOLO network model. The definitions are as follows: YOLOv4-a indicates that the SE-CSPGhostnet backbone network is employed on the basis of YOLOv4. YOLOv4-b indicates that the ASFF module is employed on the basis of YOLOv4-a. GA-YOLO indicates that an improved loss function is employed on the basis of YOLOv4-b. The comparison of the mAP and $F_1$ values of the grape detection results of the ablation experiment is displayed in Table 8. Where, $\times$ indicates that the improved module of the corresponding column is not used. Conversely, $\sqrt{}$ indicates that the improved module of the corresponding column was adopted.

In Table 8, the mAP and $F_1$ score of YOLOv4-a are 92.24% and 90.20%, respectively, which are 0.94% and 0.96% lower than that of YOLOv4, respectively. This demonstrates that after YOLOv4 is lightweight, the detection accuracy is only slightly affected. However, as displayed in Table 6, the GA-YOLO network model reduces the volume of calculation, the volume of parameter, and the weight file by 76.81%, 82.79%, and 86.68%, respectively,

compared with YOLOV4. The loss of accuracy is acceptable relative to the improvement in the volume of parameters and calculations. The mAP and $F_1$ scores of YOLOv4-b are 95.22% and 93.21%, respectively, which are 2.98% and 3.01% superior to that of YOLOv4-a, respectively. This is because the ASFF module improves the detection accuracy of the model for dense targets. ASFF enables the network to filter out contradictory and useless information, thereby retaining only useful information for combination, which solves the issue of poor detection accuracy of dense targets. The mAP and $F_1$ scores of GA-YOLO are 96.87% and 94.78%, respectively, which are 1.65% and 1.57% higher than that of YOLOv4-b, respectively. This demonstrates that application of an improved loss function improves the detection accuracy. In fact, the original loss function meets the highest accuracy conditions for the detection of 80 classes of targets in the MS COCO dataset but does not meet the highest accuracy conditions for single-target detection. Therefore, the improvement of the loss function is effective.

**Table 8.** The grape detection results of the four network models.

| Network Model | SE-CSPGhostblock | ASFF | Improved Loss Function | mAP (%) | $F_1$ |
|---|---|---|---|---|---|
| YOLOv4 | × | × | × | 93.18 | 91.16 |
| YOLOv4-a | √ | × | × | 92.24 | 90.20 |
| YOLOV4-b | √ | √ | × | 95.22 | 93.21 |
| GA-YOLO | √ | √ | √ | 96.87 | 94.78 |

### 4.3.4. Comparison of Detection Performance

In order to express the performance of the 13 models more intuitively, we draw the PR curves of the 13 network models, as depicted in Figure 12a. Where, the horizontal and vertical axes are the recall rate and the precision rate, respectively. It is evident from Formula (9) that the mAP value of the network model is the area enclosed by the PR curve and the axis of coordinates. The mAP values of the 13 network models are shown in Figure 12b. In the meantime, the parameters such as the precision rate, the recall rate, $F_1$ score and FPS of 13 network models are listed in Table 9.

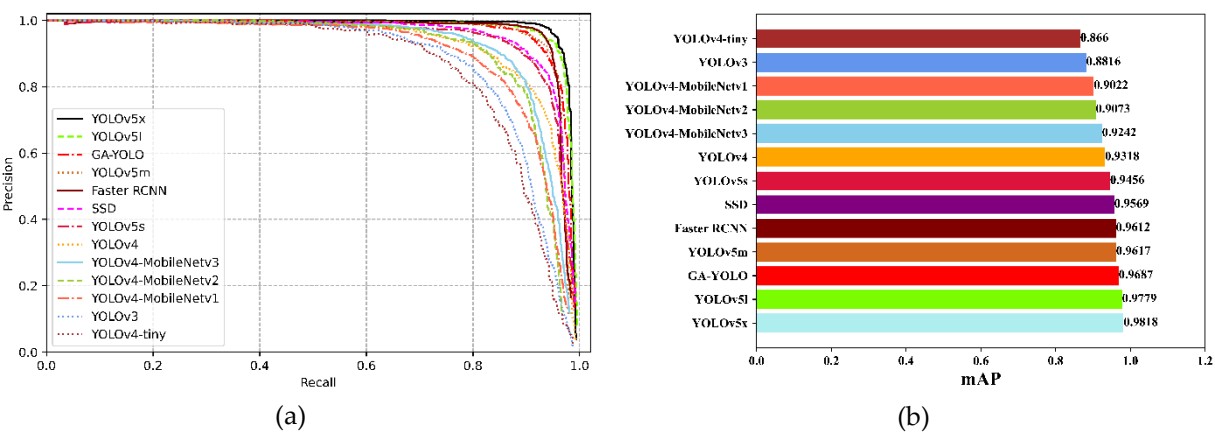

(a)    (b)

**Figure 12.** Experimental results (**a**) PR curve; (**b**) average precision chart.

In Figure 12a, the relation of the area enclosed by the PR curves of each model can be clearly seen. mAP is the average of multiple category AP, so in single object detection, it is equal to the value of AP.

In Figure 12b, the mAP of the GA-YOLO model is 96.87%, which is 0.92% and 1.31% lower than that of the YOLOv5l model and YOLOv5x model, respectively. However, according to Table 9, the detection speed of the GA-YOLO model is 55.867 FPS, which is 35.802 FPS and 43.293 FPS higher than the YOLOv5l model and YOLOv5x model, respectively. At the same time, in Table 6, the calculation volume, parameter volume, and weight file of the GA-YOLO model are 13.860 G, 11.003 M, and 32.5 M, respectively, which

are 93.62%, 87.39%, and 90.24% lower than the YOLOv5x model, respectively. Therefore, as a lightweight model, the GA-YOLO model is more appropriate for application to grape-picking robots when it comes to storage space and detection speed.

**Table 9.** Comparison of calculation volume, parameter volume, and weight file of different network models.

| Network Models | $F_1$ | Precision | Recall | FPS |
|---|---|---|---|---|
| Faster RCNN | 0.9310 | 0.9491 | 0.9136 | 10.375 |
| SSD | 0.9221 | 0.9534 | 0.8929 | 54.858 |
| YOLOv3 | 0.8570 | 0.9636 | 0.8150 | 27.254 |
| YOLOv4 | 0.9116 | 0.9426 | 0.8826 | 35.622 |
| YOLOv4-MobileNetv1 | 0.8870 | 0.9595 | 0.8247 | 33.512 |
| YOLOv4-MobileNetv2 | 0.8943 | 0.9651 | 0.8331 | 27.917 |
| YOLOv4-MobileNetv3 | 0.9023 | 0.9399 | 0.8675 | 25.859 |
| YOLOv4-tiny | 0.8495 | 0.9301 | 0.7818 | 121.374 |
| GA-YOLO | 0.9478 | 0.9533 | 0.9422 | 55.867 |
| YOLOv5s | 0.9292 | 0.9517 | 0.9078 | 44.430 |
| YOLOv5m | 0.9378 | 0.9482 | 0.9278 | 29.160 |
| YOLOv5l | 0.9550 | 0.9697 | 0.9407 | 12.574 |
| YOLOv5x | 0.9567 | 0.9675 | 0.9462 | 20.065 |

### 4.3.5. Object Detection Experiment in Actual Natural Environment

For the sake of further confirm the detection accuracy and robustness of the GA-YOLO, this paper conducts object detection experiments in the actual vineyard environment. The grape image with leaf occlusion, illumination change, and dense targets is selected for detection experiment. The grape image to be tested is shown in Figure 13.

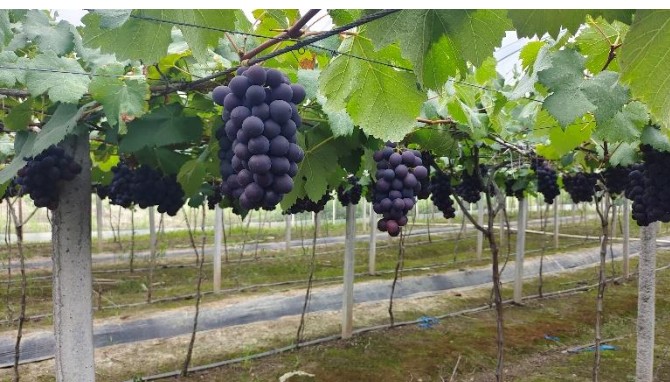

**Figure 13.** Image of grapes to be tested in the actual natural environment.

In Section 4.3.1, network models are divided into three classes according to the volume of calculation: light network models, medium network models, and large network models. The detection results of light networks, medium networks, and large networks are shown in Figures 14–16, respectively. Meanwhile, the number of grape clusters detected by 13 network models is counted as displayed in Table 10.

**Table 10.** Number of grape clusters detected by 13 network models.

| Network Models | Cluster | Network Models | Cluster |
|---|---|---|---|
| Faster RCNN | 24 | SSD | 19 |
| YOLOv3 | 16 | YOLOv4 | 17 |
| YOLOv4-tiny | 15 | GA-YOLO | 22 |
| YOLOv5s | 17 | YOLOv5m | 20 |
| YOLOv5l | 24 | YOLOv5x | 24 |
| YOLOv4-MobileNetv1 | 16 | YOLOv4-MobileNetv2 | 16 |
| YOLOv4-MobileNetv3 | 18 | | |

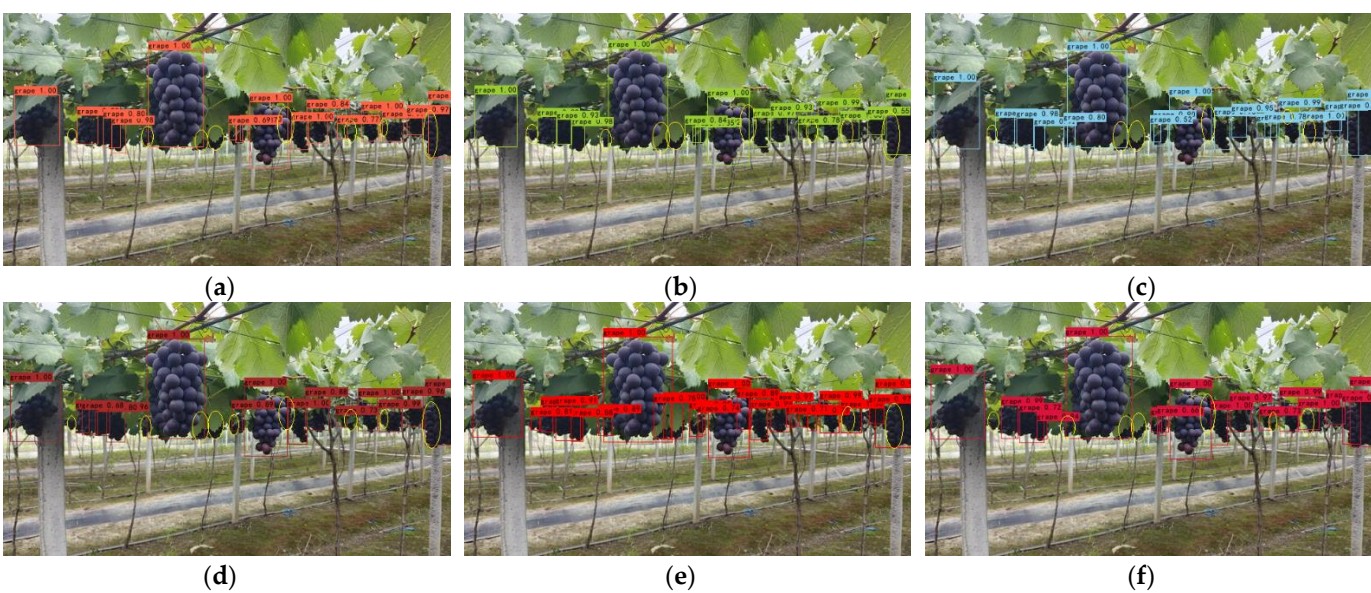

**Figure 14.** Actual grape detection results of the light network model: (**a**) YOLOv4-MobileNetv1; (**b**) YOLOv4-MobileNetv2; (**c**) YOLOv4-MobileNetv3; (**d**) YOLOv4-tiny; (**e**) GA-YOLO; (**f**) YOLOv5s. The yellow ellipse represents the missed detection of grapes.

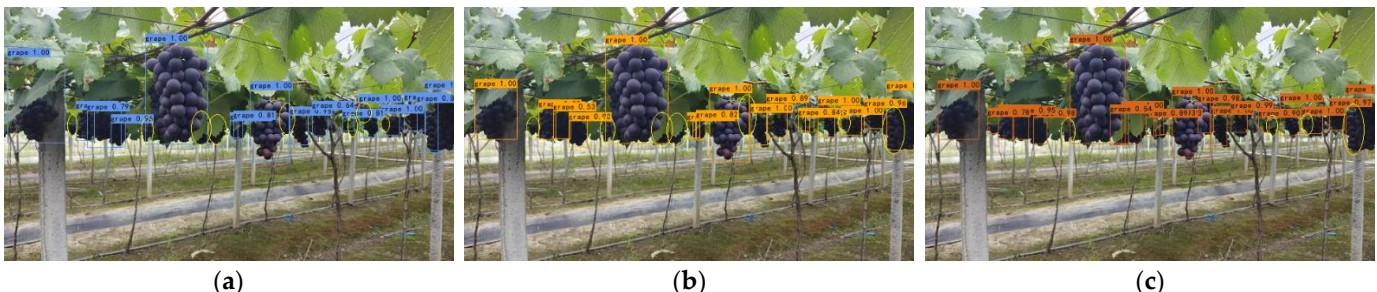

**Figure 15.** Actual detection grape results of the medium network models: (**a**) YOLOv3 (**b**) YOLOv4 (**c**) YOLOv5m. The yellow ellipse represents the missed detection of grapes.

In Figure 14, the GA-YOLO network model detects 22 clusters of grapes, which are 6 clusters, 6 clusters, 4 clusters, 7 clusters, and 5 clusters more than YOLOv4-MobileNetv1, YOLOv4-MobileNetv2, YOLOv4-MobileNetv3, YOLOv4-tiny, and YOLOv5s, respectively, indicating that it significantly outperforms other light networks in occluded and dense targets detection performance. In Figure 15, medium network models such as YOLOv3, YOLOv4, and YOLOv5m detected 16 clusters, 17 clusters, and 20 clusters of grapes, respectively. They have the phenomenon of missing detection in occluded target detection, which may be due to the defects of their feature fusion module. In Figure 16 and Table 10, large networks such as YOLOv5l and YOLOv5x detect the highest number of grape clusters, which detect 24 clusters. This is because large networks have deeper convolutional layers, which can extract richer features. At the same time, the Faster RCNN network model detects 24 clusters. Although it successfully detects most of the dense targets, it incorrectly detects the leaves as grapes in the first red oval on the left. This shows that Faster RCNN has the risk of false detection, which will reduce the picking efficiency of the robot. Among the 13 models, the GA-YOLO model can not only meet the demands of lightweight and real-time grape detection performance but also ensure the accuracy of grape detection.

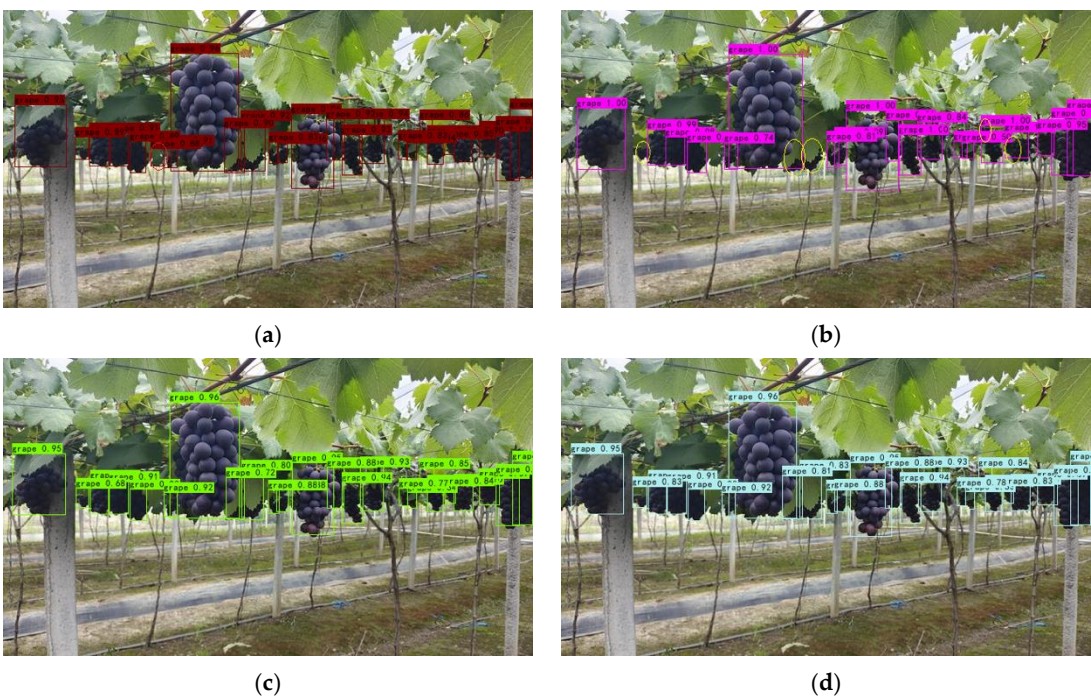

(**a**)

(**b**)

(**c**)

(**d**)

**Figure 16.** Actual grape detection results of the large network models: (**a**) Faster RCNN (**b**) SSD (**c**) YOLOv5l (**d**) YOLOv5x. The yellow circle represents the missed detection of grapes. The yellow circle represents the missed detection of grapes.

## 5. Interactive Interface

In order to make it convenient for non-professionals to use the detection model, an interactive interface based on PyQt5, as shown in Figure 17. The interface includes a detection toolbar, image detection results, and text information. There are three detection models built into the detection interface: YOLOV4, GA-YOLO, and YOLOv5s. We can choose any model for testing. The detection modes include image detection and video detection. Function buttons include start, pause, and exit systems. The text information includes four parameters: AP, FPS, precision, and recall. The running process of the whole interactive interface is divided into three steps: Step 1: Select the detection model and detection mode. Step 2: Click the Start button and call the trained weights to detect the grape targets. Step 3: save the detected results to the hard disk of the local computer.

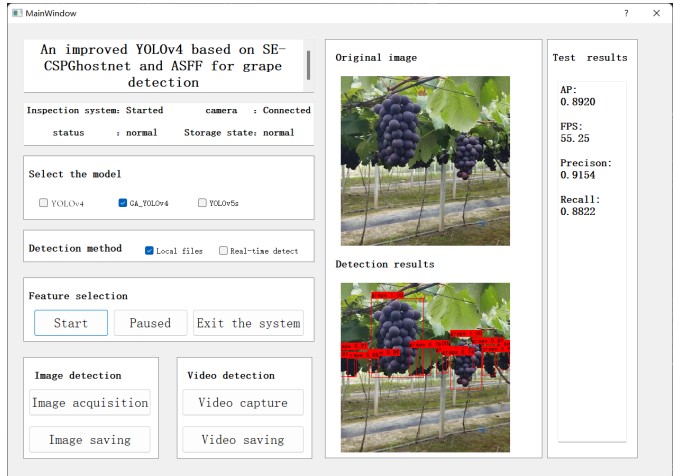

**Figure 17.** Human–machine interaction.

## 6. Discussion of Experiment

The problems of agricultural health monitoring [49–55] and harvesting [56,57] have always been hot spots of scientific research. In particular, the deep learning algorithm has become the mainstream research algorithm of the vision system of fruit-picking robots. Compared with the Faster RCNN algorithm [20], the YOLO algorithm [17–19,22] has the advantage of high speed because it unifies regression and classification into one stage. In recent years, some scholars [28–39] have applied the YOLO algorithm to the visual detection of fruit-picking robots, which provides technical help to solve the picking problem in agriculture. However, the YOLO algorithm still has some shortcomings, such as large parameters and low detection accuracy of occluded targets, which are exactly what we want to solve.

In fact, in recent years, some scholars have begun to study the lightweight model while ensuring the detection accuracy of complex objects. Zhao et al. [58] changed the backbone network in YOLOV4 from CSPdarknet53 to MobileNet53 to obtain a lightweight model, and at the same time, the deformable convolution was used to achieve dense target detection. Betti [59] and others pruned Darknet53 and compressed the backbone network from 53 layers to 20 layers. In addition, YOLO-S replaces the maximum pooling with cross-border convolution, which reduces the information loss in the transmission process and improves the detection accuracy of small targets. Huang et al. [60] proposed a GCS-YOLOv4-tiny model based on YOLOV4-Tiny. In this model, grouping convolution is used to reduce the parameter of the model by 1.7 M, and the attention mechanism is used to improve the mAP of F. margarita to 93.42%. Sun et al. [61] designed a shuffle module, lightened YOLOv5s and obtained the YOLO-P model. What is more, the YOLO-P model adopts a Hard-Swish activation function and CBAM attention mechanism. The research methods of the above scholars mainly use lightweight modules to partially replace the original network to achieve the purpose of reducing parameters. Meanwhile, methods such as the replacement of activation functions and the addition of attention mechanisms ensure the detection accuracy of the model for occlusions and dense objects. To conclude, using lightweight convolution modules (such as depth separable convolution, group convolution, etc.), and replacing backbone networks are the most frequently used to reduce the parameters of the model. Mao et al. [32], Fu et al. [35], Li et al. [36], and Liu et al. [37] used depth separable convolution to reduce the parameters of the model by 77%, 18.18%, 49.15%, and 10.03%, respectively. Moreover, Zhang et al. [33], Cui et al. [38], Zeng et al. [39], and Zhao et al. [58] replaced the backbone network to reduce the parameters of the model by 82.3%, 52.3%, 78%, and 82.81%, respectively. Replacing backbone networks can reduce more parameters than using lightweight convolution modules, but the accuracy drops even more. Similar to using deep separable convolution to replace ordinary convolution, we use a ghost module to replace ordinary convolution, which reduces the parameters of the model by 82.79%, and the accuracy loss is less affected than replacing the backbone network. In order to solve the problem of decreasing accuracy, attention mechanisms and improving loss function are common methods, which have been adopted by most researchers [26,27,33,34,36–38,60,61]. In addition to these two improved methods, we adopt the ASFF method [42] in the head network to effectively improve the detection accuracy of the model. ASFF performs spatial filtering on the feature maps at all levels, thus retaining only useful information for combination. GA-YOLO is proposed under the guidance of similar design ideas. The GA-YOLO model is of great significance for improving the picking speed and picking quality (low missing picking rate) of the picking robot.

The model proposed in this paper mainly aims at the target detection of dense and occluded grapes. The model can also be used for other fruits in the same growth state (clusters) such as tomatoes, bananas, and strawberries. According to the ablation experiments in Section 4, we found that the detection accuracy of the model decreased by 0.94% after the model was lightened by 82.79%. Yet, we can add an ASFF module and improve the loss function to heighten accuracy. The model is lightweight, which is of great significance to solve the deployment problem of the mobile end of the model. In addition, the recall

rate of the GA-YOLO model and other target detection models is lower than the precision rate, which shows that the problem of missed detection is puzzling grape detection. By lowering the confidence threshold for prediction, it is easier for the model to detect grapes and reduce the missed detection rate. However, this will increase the risk of false detection, so subsequent debugging of the model is required. Finally, it may be possible to increase the size of the input image to obtain more abundant location features and semantic features to reduce the missed detection rate, but this method will increase the number of parameters of the model, so it is necessary to find the optimal input image size.

There are still some problems to be considered when picking grapes by picking robots.

(1) We need to distinguish the maturity of grapes to avoid picking immature grapes.
(2) The detection of grape clusters is only a part of picking steps, and we also need to realize the detection of picking points. Some scholars [62–64] have developed the detection algorithm of grape picking points, the position errors between most predicted picking points and real points are within 40 pixels. However, the detected grapes are not in dense and shaded conditions, and the detection accuracy is low, so there is much room for improvement. The occlusion problem is not only solved by visual models but also requires appropriate planting strategies, such as farmers paying attention to thinning leaves and fruits when planting.
(3) The picking robot can work 24 h a day, so it is necessary to obtain the grape dataset at night. In fact, when the fruit is picked at night, it will not be exposed to the sun to cause water loss, so the quality of the fruit will be better. In addition, a richer dataset can increase the robustness of GA-YOLO.

Overall, the development of deep learning-based methods for fruit detection in agricultural settings has shown great promise in recent years. Other deep learning models (such as Faster RCNN and SSD, etc.), have their own unique advantages in specific fruit detection. In the future, we can also combine the design ideas of these models to better solve the identification problem of tropical fruits.

## 7. Conclusions

The goal of this paper is to decrease the parameters and calculations and raise the detection accuracy of the model. In this research, a lightweight network model named GA-YOLO was proposed. This model uses a backbone network of SE-CSPGhost, which reduces the parameter amount of the original model by 82.79% and improves the detection speed of the model by 20.245 FPS. This lightweight approach is of great significance for model deployment to mobile terminals. At the same time, although the lightweight model reduces the detection accuracy of dense and occluded grapes by 0.94%. By adding the attention mechanism and ASFF mechanism, and improving the loss function, the accuracy rate is increased by 3.69%. In short, the parameter quantity of the GA-YOLO model is 11.003 M, the mAP is 96.87%, the detection speed is 20.245 FPS and the $F_1$ value is 94.78%. Compared with YOLOv4 and the other 11 commonly used models, the GA-YOLO has the advantages of high detection accuracy and low model parameters. It has excellent comprehensive performance and can meet the precision and speed requirements of picking robots. Finally, we use PyQt5 to design a human–computer interaction interface to facilitate the use of the GA-YOLO model by non-professionals. In future research, we will consider the mobile deployment of the model, and deploy the GA-YOLO model on small computing devices (Raspberry Pie, developed by the Raspberry Pie Foundation in Cambridge, England; Jetson Nano, developed by the NVIDIA Corporation in Santa Clara, CA, USA; Intel NCS 2, developed by the Intel Corp in Santa Clara, CA, USA), using the deep learning inference framework NCNN and TensorRT. In addition, we will consider collecting grape datasets under night illumination and training a widely used GA-YOLO model.

**Author Contributions:** Conceptualization, J.C. and A.M.; methodology, A.M. and L.H.; software, L.H. and A.M.; validation, Y.S.; formal analysis, W.L.; investigation, H.Z.; resources, J.C.; data curation, Z.W.; writing—original draft preparation, A.M.; writing—review and editing, A.M. and L.H; visualization, A.M. and L.H.; supervision, J.C.; project administration, A.M.; funding acquisition, J.C. All authors have read and agreed to the published version of the manuscript.

**Funding:** This research was funded by the National Nature Science Foundation of China (grant number 62163005), the Natural Science Foundation of Guangxi Province (grant number 2022GXNSFAA035633).

**Institutional Review Board Statement:** Not applicable.

**Informed Consent Statement:** Not applicable.

**Data Availability Statement:** Not applicable.

**Conflicts of Interest:** The authors affirm that they have no known financial or interpersonal conflict that would have appeared to have an impact on the research presented in this study.

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
