# Peer review of "GA-YOLO: A Lightweight YOLO Model for Dense and Occluded Grape Target Detection"

_horticulturae, doi:10.3390/horticulturae9040443_

Round 1

Reviewer 1 Report

This paper shows a theme smart agriculture. The contribution is significant to the advancement of knowledge. However, some points need to be better detailed for a complete understanding.

Title: ok.

Abstract: Suggestion to include the study region/location and year.

Introduction: is comprehensive whit a good overview of problem and context. Write more clearly the main objective of the study in the final part of the Introduction.

Methodologies: the method description is good. However, there was a doubt about using only two cited references/studies for a relatively complex analysis.

Results and Discussion: There is no discussion with references to previous studies in the text. "Authors should discuss the results and how they can be interpreted in perspective of previous studies and of the working hypotheses. The findings and their implications should be discussed in the broadest context possible and limitations of the work highlighted. Future research directions may also be mentioned. This section may be combined with Results. (https://www.mdpi.com/journal/horticulturae/instructions)

Conclusions: It should focus on the main results obtained. Add the "future work" text in the same item.

Obs. The format submission presented is in the https://www.mdpi.com/journal/agriengineering

Author Response

Dear Editors,

The attachment is  is a cover letter to explain, point by point, the details of the revisions to the manuscript and my responses to your comments.
Thank you and best regards.

Yours sincerely,

Aoqiang Ma

Name: Aoqiang Ma

Address: College of Mechatronic Engineering, Guangxi University, Nanning

E-mail: aoqiangma@163.com

Reviewer 2 Report

Your work is interesting, written well, and organized. This manuscript reports on a study of GA-YOLO: a lightweight YOLO model for dense and occluded grape target detection. The study design meets the general standards and from what I can judge the data is being collected and analyzed appropriately. This work is an unpublished manuscript with relevant information that should be made public in a scientific journal for discussion among scientists working in the field.

However, some comments should be considered before publishing, in this way, the social and scientific relevance of the manuscript would be improved:

Line 9: should say: Picking robots

Line 10: to realizing robot

Line 15: of the model

Line 29, 30, 31: labor

Line 40: the detection of the grape region

Line 40: delete so as to. Should say: to realize

Line 45: the background

Line 51: a specified

Line 50: greens grapes

Line 53: artificial features

Line 72: a large number

Line 77: the convolution module and reducing the convolution

Line 93: lightweight,

Line 101: improve detection

Line 118: the basic

Line 134: uses labellmg

Line 141: Labellmg software

Line 169: a pooling

Line 199: the convolution kernel

Line 231: weight is

- Discussion

The discussion of results is missing in the scientific article, which does not show the real contribution to knowledge in the field of knowledge in which a topic is studied, investigated, or tries to solve a specific situation. It must maintain scientific and methodological rigor.

Line 482: I continue to add a paragraph that summarizes the importance, usefulness, and social relevance, contemporary of the study, specifically pointing out the Impact, Benefit, and Projection, something like this (for example):

The scientific literature establishes that the YOLO algorithm is one of the fastest and has already been used in fruit-picking robots [43] due to the results of the comparison of the standard Faster R-CNN algorithm, their proposed modification of Faster R -CNN and YOLOv3 for the detection of oranges, apples, and mangoes.

When comparing the accuracy of the proposed architecture with similar studies found in the literature, it is found that Lee et al. [44], obtained an accuracy of 99%; in this study, the authors performed an adjustment on the images, removing the background and leaving only the potato leaf in the image for subsequent CNN training; on the other hand, Islam et al. [45] obtain an accuracy of 99.43% but use transfer learning to train the CNN, proposed by the authors, using VGG16. Likewise, the research by Rey et al. [46], Campos et al. [47], Vega et al. [48], and Orlando [49] used machine learning algorithms such as Random Forest and regularized optimal scaling regression to accurately identify soil properties associated with symptoms of tropical diseases in bananas. Therefore, it is possible to effectively use the lightweight YOLO model for dense and occluded grape target detection, proposed architecture from images, and provide highly accurate results, due to the size of the architecture and the number of parameters it comprises. These characteristics make the proposed architecture suitable for the creation of various types of human-machine interfaces.

When considering machine learning studies in tropical fruits such as bananas [48, 49, 50], the use of deep learning techniques for fruit detection has also been investigated. For example, a recent study developed a deep learning-based approach for the detection of banana bunches in images captured in the field. The approach utilized a Faster R-CNN (Region-based Convolutional Neural Network) architecture and achieved high accuracy in detecting banana bunches, even in occluded environments [51].

Overall, the development of deep learning-based approaches for fruit detection in agricultural environments has shown great promise in recent years. The GA-YOLO model, in particular, is an effective solution for the detection of dense and occluded grape targets, while other deep learning models like Faster R-CNN are effective for the detection of other tropical fruits like bananas. Further research in this area is needed to improve the accuracy and robustness of these models for broader use in agricultural settings.

Our results are an interesting contribution to the knowledge focused on the fact that precision agriculture advances hand in hand with images and indices that allow the producer to be guided in making more efficient decisions about crops and agricultural tasks.

References

I suggest adding recent references which address the issue in question in Latin American territories. I suggest incorporating the new recommended references in the discussion section, which would improve the scientific quality of the candidate manuscript. Suggested citations are for genuine scientific reasons that emphasize the current topic of study in context:

43. Kuznetsova A, Maleva T, Soloviev V. Detecting Apples in Orchards Using YOLOv3 and YOLOv5 in General and Close-Up Images. In: Han, M., Qin, S., Zhang, N. (eds) Advances in Neural Networks – ISNN 2020. ISNN 2020. Springer, Cham, Lecture Notes in Computer Science, 2020, (12557). https://doi.org/10.1007/978-3-030-64221-1_20

44. Lee TY, Yu JY, Chang YC, Yang JM. HealthDetection for Potato Leaf with Convolutional NeuralNetwork. Indo - Taiwan 2nd International Conference on Computing, Analytics and Networks, Indo-Taiwan ICAN2020 – Proceedings. p.289–293, 2020. https://doi.org/10.1109/Indo-TaiwanICAN48429.2020.9181312

45. Islam F, Hoq MN, Rahman CM. Application of transfer learning to detect potato disease from leaf image. IEEE International Conference on Robotics, Automation, Artificial-Intelligence and Internet-of-Things (RAAICON). p.127-130, 2019, https://doi.org/10.1109/RAAICON48939.2019.53

46. Rey JC, Orlando B, Lobo D, Navas-Cortés JA, Gómez JA, Landa BB. Fusarium Wilt of Bananas: A Review of Agro-Environmental Factors in the Venezuelan Production System Affecting Its Development. Agronomy, 2021, 11(5):986, https://doi.org/10.3390/agronomy11050986

47. Campos, O. F. Paredes, J. Rey, D. Lobo, S. Galvis-Causil, The relationship between the normalized difference vegetation index, rainfall, and potential evapotranspiration in a banana plantation of Venezuela, STJSSA, 2021, 18(1): 58-64, http://dx.doi.org/10.20961/stjssa.v18i1.50379

48. Vega, A.; Orlando, B.O.; Rueda Calderón, M.A.; Montenegro-Gracia, E.; Araya-Almán, M.; Marys, E. Prediction of Banana Production Using Epidemiological Parameters of Black Sigatoka: An Application with Random Forest. Sustainability 2022, 14, 14123. https://doi.org/10.3390/su142114123

49. Orlando O, Calero J, Rey JC, Lobo D, Landa BB, Gómez JA. Correlation of banana productivity levels and soil morphological properties using regularized optimal scaling regression. Catena, 2022, 208: 105718. https://doi.org/10.1016/j.catena.2021.105718

50. Rey, J.C., Campos, O., Perichi, G., Lobo, D. Relationship of Microbial Activity with Soil Properties in Banana Plantations in Venezuela. Sustainability 2022, 14, 13531. https://doi.org/10.3390/su142013531

51. Le, T. T., Lin, C. Y. Deep learning for noninvasive classification of clustered horticultural crops–A case for banana fruit tiers. Postharvest Biology and Technology, 2019, 156, 110922. https://doi.org/10.1016/j.postharvbio.2019.05.023

Author Response

Dear Editors,

The attachment is a cover letter to explain, point by point, the details of the revisions to the manuscript and my responses to your comments.

Thank you and best regards.

Yours sincerely,

Aoqiang Ma

Name: Aoqiang Ma

Address: College of Mechatronic Engineering, Guangxi University, Nanning

E-mail: aoqiangma@163.com

Round 2

Reviewer 1 Report

The authors have improved the text. However, the discussions can be more detailed based on the references included in the new version of the paper.

Author Response

Dear reviewer1:

The attachment is a "Cover letter to reviewer1" is a cover letter to explain, point by point, the details of the revisions to the manuscript and my responses to your comments.

Thank you and best regards.

Yours sincerely,

Aoqiang Ma
